# PredNTS: Improved and Robust Prediction of Nitrotyrosine Sites by Integrating Multiple Sequence Features

**DOI:** 10.3390/ijms22052704

**Published:** 2021-03-08

**Authors:** Andi Nur Nilamyani, Firda Nurul Auliah, Mohammad Ali Moni, Watshara Shoombuatong, Md Mehedi Hasan, Hiroyuki Kurata

**Affiliations:** 1Department of Bioscience and Bioinformatics, Kyushu Institute of Technology, 680-4 Kawazu, Iizuka, Fukuoka 820-8502, Japan; nurnilamyani.514@gmail.com (A.N.N.); Firdana.525@gmail.com (F.N.A.); 2WHO Collaborating Centre on eHealth, UNSW Digital Health, School of Public Health and Community Medicine, Faculty of Medicine, UNSW Sydney, Sydney, NSW 2052, Australia; m.moni@unsw.edu.au; 3Center of Data Mining and Biomedical Informatics, Faculty of Medical Technology, Mahidol University, Bangkok 10700, Thailand; watshara.sho@mahidol.ac.th; 4Japan Society for the Promotion of Science, 5-3-1 Kojimachi, Chiyoda-ku, Tokyo 102-0083, Japan

**Keywords:** nitrotyrosine, post-translational modification, feature encoding, RFE feature selection, machine learning

## Abstract

Nitrotyrosine, which is generated by numerous reactive nitrogen species, is a type of protein post-translational modification. Identification of site-specific nitration modification on tyrosine is a prerequisite to understanding the molecular function of nitrated proteins. Thanks to the progress of machine learning, computational prediction can play a vital role before the biological experimentation. Herein, we developed a computational predictor PredNTS by integrating multiple sequence features including K-mer, composition of k-spaced amino acid pairs (CKSAAP), AAindex, and binary encoding schemes. The important features were selected by the recursive feature elimination approach using a random forest classifier. Finally, we linearly combined the successive random forest (RF) probability scores generated by the different, single encoding-employing RF models. The resultant PredNTS predictor achieved an area under a curve (AUC) of 0.910 using five-fold cross validation. It outperformed the existing predictors on a comprehensive and independent dataset. Furthermore, we investigated several machine learning algorithms to demonstrate the superiority of the employed RF algorithm. The PredNTS is a useful computational resource for the prediction of nitrotyrosine sites. The web-application with the curated datasets of the PredNTS is publicly available.

## 1. Introduction

Nitrotyrosine, which is generated by numerous reactive nitrogen species, is a type of protein post-translational modification. It occurs when tyrosine is nitrated by reactive nitrogen species (RNS) such as peroxynitrite anions (ONOO^−^), which are carried out in vivo through the rapid reaction of nitric oxide (NO^−^) and superoxide (O_2_^−^) [1,2,3]. Nitration of proteins changes their chemical properties. Excessive levels of peroxynitrite anion are observed during the inflammation process in a large number of diseases including diabetics, cancer, neurodegenerative disorders, asthma, and ageing [4,5].

Tyrosine nitration occurs within a protein interaction region such as enzyme–substrate or receptor–ligand complexes and brings several effects, like decreasing the electron intensity of the phenolic ring of tyrosine, and negatively affects their interactions. It modifies the enzymes and receptors, reducing their biological activities [2,6]. Besides, tyrosine nitration may interfere directly with the phosphorylation of tyrosine residues responsible for cellular signalling pathways [7,8]. Therefore, potential guidance for developing new therapeutic strategies and drugs can be obtained through the identification of site-specific nitration modifications on tyrosine.

To reveal the mechanism and function of nitroprotein, identification of potential nitrotyrosine sites is essential. To date, large-scale proteomic studies have been performed to identify nitrated proteins based on the molecular signature of nitrotyrosine sites [3,5,8]. Although the number of experimentally verified nitrotyrosine sites is increasing, the mechanism of site-specific nitration modification on tyrosine remains largely unknown [9,10,11], probably owing to technological measurement constraints. The conventional experimental methods provide insights into a biological role of nitrotyrosine sites, but they are very time-consuming and expensive. Therefore, as an alternative strategy, an in silico approach can be proposed to predict nitrotyrosine sites, which functions for all proteome annotations because of their efficiency and convenience.

Until now, only a few predictors have been presented to identify nitrotyrosine sites [12,13,14,15]. Initially, Liu et al. [15] developed a predictor based on a group-based prediction system (GPS) using four statistical procedures (i.e., selection of motif length, K-means clustering, matrix mutation, and weight training), named GPS-YNO2. Xie et al. [12] developed the DeepNitro that implemented deep learning with four encoding schemes (i.e., positional amino acid distributions, sequence contextual dependencies, physicochemical properties, and position-specific scoring features). The NTyroSite [13] was constructed using sequence evolutionary information. The iNitro-Tyr was developed based on the pseudo amino acid composition [14]. From the results of previous studies, it is known that the previous predictors use training datasets and receive good performance by five-fold cross-validation (CV) tests. In this study, we have developed a computational predictor PredNTS by integrating multiple sequence features of K-mer, composition of k-spaced amino acid pairs (CKSAAP), AAindex, and binary. A workflow of the PredNTS is shown in Figure 1. We implemented the recursive feature elimination (RFE) to select the important features via a random forest (RF) classifier. Finally, the successive RF scores were combined with a linear regression model. A user-friendly web server was developed and is freely available at http://kurata14.bio.kyutech.ac.jp/PredNTS/ (accessed on 1 March 2021).

## 2. Results and Discussion

### 2.1. Sequence Preference Analysis

We visualized the curated positive and negative samples using a graphical sequence logo to check the significant preference of the amino acid residues surrounding nitrotyrosine proteins [16], as shown in Figure 2. Some significant differences of amino acid sequences were detected between the positive and negative samples. We found that charged residues such as K, R, and E frequently appeared in the enriched positions, while Y, S, F, and L were frequently observed in the depleted section. However, in the depleted section, no stacked residue was found at positions of −16, +3, and +13. The above analysis of amino acid residue preference between the positives and negatives suggested that a combination of frequency-based encodings with position specific encodings is effective in designing nitrotyrosine site prediction.

### 2.2. Single Encoding-Employing RF Model on the Training Dataset

We used the four encoding schemes (AAIndex, binary, CKSAAP, and K-mer) to generate numerical feature vectors. The window size was set to 41 (−/+20) for all the encoding schemes. The prediction performances were measured using five-fold CV through the RF classifier. The average performance of the four single encoding-employing RF models without any feature selection is summarized in Table 1. Without any feature selection, the K-mer encoding performed better than any other encodings, which achieved Ac of 0.796 and Matthew’s correlation coefficient (MCC) of 0.593.

Note that the high-dimensional features may contain irrelevant or redundant attributes that affect accuracy reduction [17,18]. To discriminate the relative contribution and importance of each feature, the RFE method was considered. Different feature subsets were selected for each encoding, which controlled the high ranked features ranging from the top 50 to all with an interval of 50. The curated subset features were inputted to RF separately and their individual performances were estimated using five-fold CV (Appendix A). This approach selected the 400- for AAindex, 300- for binary, 200- for CKSAAP, and 500-dimesional features for K-mer. Then, we measured the four statistical measures of Sp, Sn, Ac, and MCC via a five-fold CV test on the training dataset, as shown in Table 2. Use of the RFE improved the performance of our models. In the model with feature selection, the Acc was ~2% to 4% higher than the model without any feature selection. Figure 3A shows the ROC curves for the four single encoding-employing models with feature selection on the training datasets. The CKSAAP and K-mer encodings provided better prediction than the other two encoding schemes. The CKSAAP and K-mer encodings achieved AUCs of 0.900 and 0.895, respectively, while the binary and AAindex encodings provided AUCs of 0.773 and 0.771, respectively. The feature selection improved the performance for all the encoding schemes (binary, AAindex, CKSAAP, and K-mer).

### 2.3. Single Encoding-Employing RF Model on the Independent Dataset

We used an independent dataset to investigate the robustness of the training models. The performances of the models without any feature selection and the models with feature selection were evaluated on the independent dataset, as shown in Table 3 and Table 4, respectively. The use of RFE improved the prediction performance of nitrotyrosine sites. As shown in Figure 3B, the CKSAAP and K-mer encodings achieved AUCs of 0.833 and 0.857, respectively, while the binary and AAindex encodings provided AUCs of 0.720 and 0.750, respectively. The CKSAAP and K-mer encodings performed better than the binary and AAindex encodings.

### 2.4. Prediction Performance of PredNTS

To build the PredNTS, we linearly combined the probability scores generated by the four types of single encoding-employing RF models. We optimized the weight coefficients for the binary, AAindex, CKSAAP, and K-mer encodings as 0.01, 0.01, 0.3, and 0.68, respectively. As shown in Table 5, the PredNTS achieved an AUC of 0.910 on the training dataset by five-fold CV, while it achieved an AUC of 0.860 on the independent dataset. The integration of the four encoding schemes greatly improved the prediction performance. To validate the superiority of the RF employed by the PredNTS, we compared it with the two machine learning algorithms of naïve Bayes (NB) and k-nearest neighbor (KNN). Here, we employed the same number of selected features and the same window size of 41. The performances of the three machine learning algorithms were compared for the combined models without and with feature selection, respectively, as shown in Figure 4A,B. The AUCs of the PredNTS were 3–6% higher than those of the NB and KNN implementing combined models. The RF outperformed the NB and KNN, demonstrating the superiority of the RF.

### 2.5. Comparison of PredNTS with Other Existing Predictors

Several computational predictors for nitrotyrosine sites have been developed. We compared the PredNTS predictor with the three existing predictors: GPS-YNO2 [14], DeepNitro [12], and NTyroSite [15]. The comparison was carried using the independent dataset with 203 positive samples and 1022 negative samples. The curated independent datasets were submitted to the GPS-YNO2, DeepNitro, and NTyroSite online servers, then the performance was assessed by the four statistical measures (Sn, Sp, Acc, and MCC). As seen in Table 6, our predictor presented much better performance than the GPS-YNO, DeepNitro, and NTyroSite predictors in terms of Sn, Sp, Ac, and MCC. The PredNTS achieved 0.522 for Sn, 0.809 for Sp, 0.761 for Acc, and 0.286 for MCC. The PredNTS presented significantly higher MCC than the other predictors. This might be caused by the fact that the GPS-YNO2 did not use any independent dataset to evaluate its robustness and that the DeepNitro and NTyroSite did not integrate their encoding schemes.

### 2.6. Web Server Implementation

The PredNTS webserver was developed to serve potential user communities and is freely available at http://kurata14.bio.kyutech.ac.jp/PredNTS/ (accessed on 1 March 2021). On the main page, users submit a query protein sequence by pasting it into the text box or using the browse button. The server initially calculates the window size based on the number of tyrosine residues. In the meantime, the server generates the PSSM by performing the PSI-BLAST search for the query sequence and encodes the sequence windows. After selecting the important features using the RFE method, the server classifies the feature vectors using the RF algorithm. The webserver returns the predicted results containing the residue positions, request protein name, job ID, and probability scores on the output webpage. The server creates a job ID such as “20200102100011”. Users can save this ID for the future enquiry. The PredNTS implementation can handle only FASTA format sequences.

## 3. Materials and Methods

### 3.1. Dataset Construction

We collected the datasets from the different public sources including DeepNitro and iNitro-Tyr [12,14]. The experimentally identified nitrotyrosine sites (“Y”, tyrosine residue) were considered as positive samples, whereas the resting Y residues were measured as negative samples [19,20,21]. It contains 796 nitrotyrosine proteins with 1406 experimentally validated nitrotyrosine sites. A sequence window with a length of 2w + 1 was prepared so as to place nitrotyrosine in the centre. We removed redundant sequences by considering a threshold of 40% level by CD-HIT [22]. Finally, randomly selected 20% of the samples (203 positive samples and 1022 negative samples) were considered as the independent dataset to examine the model strength. From the whole remaining dataset, we pooled a 1:1 ratio of positive to negative samples (1191 positive samples and 1191 negative samples) as the training model to avoid possible biased predictions. The independent (203 positive samples and 1022 negative samples) dataset was used to compare the proposed PredNTS model with existing predictors.

### 3.2. Sequence Encoding Scheme

The binary amino acid encoding scheme was used to encode position information from the sequence windows [23,24,25]. Here, by adopting the binary encoding, we converted a 41 amino acid sequence, including the gap that is represented as (-), into a 861 (=41 × 21)-dimensional feature vector.

The physicochemical properties of amino acids have been extracted from the AAindex database 24 (version 9.1) [26]. Herein, we used 15 types of AAindex properties to generate a 615 (=41 × 15)-dimensional vector.

The composition of k-spaced amino acid pairs (CKSAAP) encoding is the composition of the k-spaced residue pairs in the window, which is widely used in a protein bioinformatics field [14,20,27]. In this scheme, *k* represents the gap length of two amino acids. For example, *k* = 0 provides 400 amino acid residue pairs (i.e., AA, AC, AD,..., YY). At *k* = 0, 1, 2, 3, and 4, it generates a 2000-dimensional feature vector. Details of the CKSAAP encoding are described in our previous studies [14,25].

The K-mer encoding is widely used in the field of genomics and bioinformatics [23,28,29,30,31]. We employed the K-mer to minimize the impact of an arbitrary starting point. The K-mer encodes a monopeptide into a 20-dimensional feature vector at K = 1. Similarly, at K = 2 and 3, it encodes dipeptides and tripeptides, which generates an 8020-dimensional feature vector.

### 3.3. Feature Selection

We considered the RFE as a feature selection approach to remove non-essential features from the dataset [32]. This method was classified as a wrapper method, which started from building a learning model for the entire dataset. We calculated the important scores from each predictor and trimmed the least important features out of the current set of features. The procedure is repeated until the number of optimal performance features converges. The ‘rfe’ function from the ‘Caret’ R package was adopted to obtain the important features.

### 3.4. Machine Learning Algorithm

The RF is a supervised and ensemble machine learning classifier that combines multiple tree-based representations to create a more powerful and interpretable model. It is widely used in protein bioinformatics research [33,34,35,36,37,38,39,40,41,42,43]. It performs as a huge assortment of uncorrelated decision trees, and the votes are carried to decide the final classification from the whole trees. The prediction model of our PredNTS was built using the ‘RandomForest’ R package (https://cran.r-project.org/web/packages/randomForest/ (accessed on 1 March 2021)). In addition, we compared the RF algorithm with the naïve Bayes (NB) and k-nearest neighbor (KNN) algorithms. An R package of an NB algorithm (https://cran.r-project.org/web/packages/naivebayes/ (accessed on 1 March 2021)) was employed to classify the nitrotyrosine proteins, while the R package (https://rpubs.com/njvijay/16444 (accessed on 1 March 2021)) was used to build the KNN model.

### 3.5. Evaluation Measure

Our PredNTS model is evaluated through five-fold CV. This method starts from randomly selecting and partitioning the training dataset into five sub-folds, then those five sub-folds are divided into the training and test datasets alternately. Four folds are used as the training sets; the remaining one fold is used as the test set. This process is repeated five times in order to measure the entire dataset. Four simple statistical measures of specificity (Sp), sensitivity (Sn), accuracy (Acc), and Matthew’s correlation coefficient (MCC) [44,45,46,47,48,49,50,51,52,53,54,55,56,57,58,59] are considered to evaluate the prediction performance of the model, as follows:(1)$$Sn=\frac{TP}{TP+FN}$$
(2)$$Sp=\frac{TN}{TN+FP}$$
(3)$$Ac=\frac{TP+TN}{TP+TN+FP+FN}$$
(4)$$MCC=\frac{\left(TP\times TN\right)-\left(FP\times FN\right)}{\sqrt{\left[TP+FN\right]\left[TN+FP\right]\left[TP+FP\right]\left[TN+FN\right]}}$$
where *TP* is the true positive, *FP* is the false positive, *TN* is the true negative, and *FN* is the false negative. The statistical values of (Sn, Sp, Acc) are between 0 and 1 and MCC is −1 to 1. Prediction accuracy is indicated by a high score. In addition, to obtain the area under the curve (AUC), we plot the receiver operating characteristics (ROC) curve, which is used to measure the overall ability of a classifier; the pROC package in the R language (https://cran.r-project.org/web/packages/pROC/ (accessed on 1 March 2021)) is used for this process.

## 4. Conclusions

We have developed a computational predictor PredNTS that linearly combined the probability scores generated by multiple single encoding-employing RF models. The critical features were selected by the RFE approach using RF. The employed RF algorithm was shown to be superior to other machine learning algorithms. On both the training and independent datasets, the PredNTS achieved excellent prediction performances, outperforming existing state-of-the-art predictors. The PredNTS is a useful computational resource for the prediction of nitrotyrosine sites, and is freely available at http://kurata14.bio.kyutech.ac.jp/PredNTS/ (accessed on 1 March 2021).

## Figures and Tables

**Figure 1 ijms-22-02704-f001:**
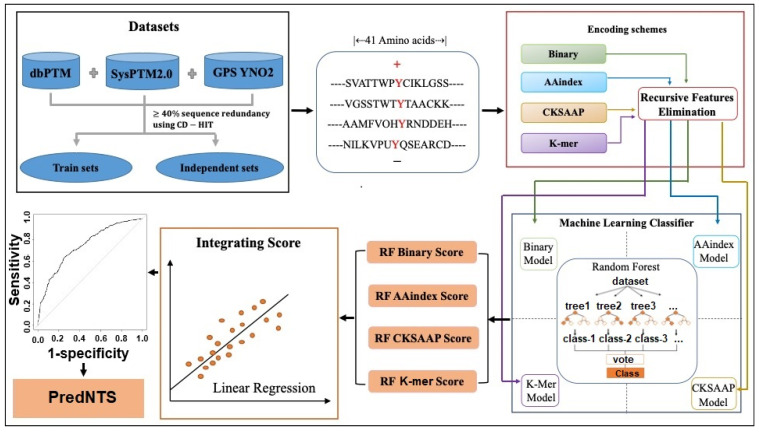
A framework for the PredNTS predictor and its server application. GPS, group-based prediction system; RF, random forest; RFE, recursive feature elimination; CKSAAP, composition of k-spaced amino acid pairs.

**Figure 2 ijms-22-02704-f002:**
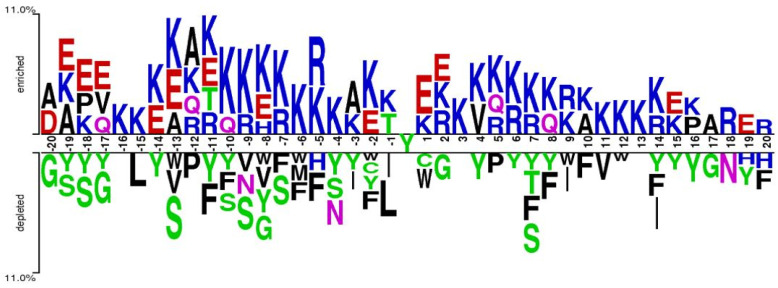
Sequence preference analysis of positive and negative samples of nitroproteins.

**Figure 3 ijms-22-02704-f003:**
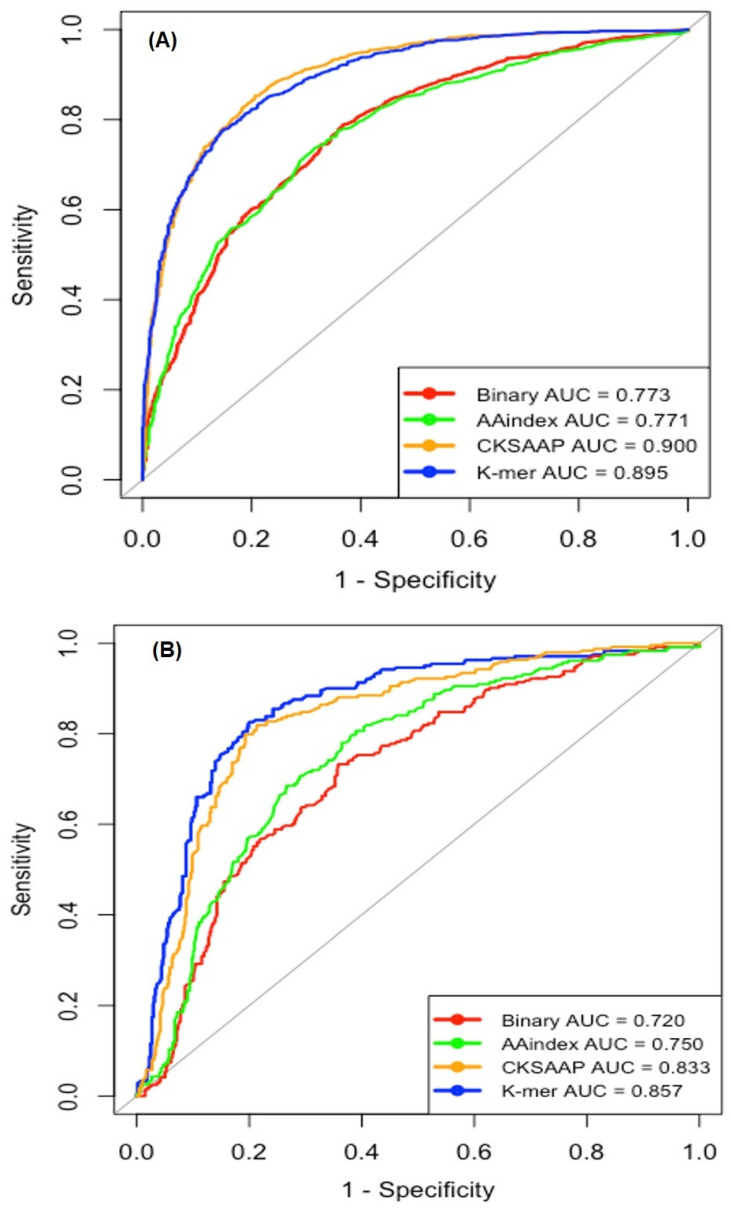
Receiver operating characteristics (ROC) curves of four types of the single encoding-employing models. (**A**) On the training dataset using five-fold cross-validation test. (**B**) On the independent test dataset. AUC, area under the curve.

**Figure 4 ijms-22-02704-f004:**
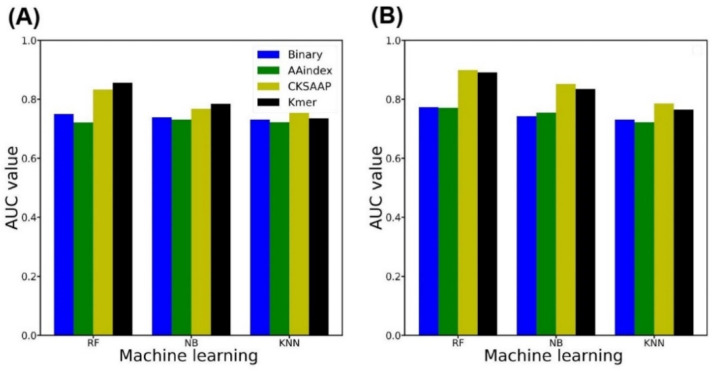
AUC values of the single encoding-employing models with different machine learning algorithms on training datasets. (**A**) Performance comparison of the models without any feature selection. (**B**) Performance comparison of the models with feature selection. NB, naïve Bayes; KNN, k-nearest neighbor; RF, random forest.

**Table 1 ijms-22-02704-t001:** Prediction performance of the single encoding-employing models without any feature selection on the training dataset by five-fold cross-validation (CV). CKSAAP, composition of k-spaced amino acid pairs. MCC, Matthew’s correlation coefficient.

Encoding Scheme	Sn	Sp	Acc	MCC
Binary	0.519 (0.19)	0.808 (0.01)	0.658 (0.04)	0.331 (0.14)
AAindex	0.473 (0.24)	0.803 (0.00)	0.638 (0.02)	0.293 (0.16)
CKSAAP	0.731 (0.16)	0.808 (0.00)	0.709 (0.11)	0.519 (0.20)
K-mer	0.785 (0.11)	0.807 (0.01)	0.796 (0.06)	0.593 (0.13)

Values in parentheses represent the standard error (SE).

**Table 2 ijms-22-02704-t002:** Prediction performance of the single encoding-employing models with feature selection on the training dataset by five-fold CV.

Encoding Scheme	Sn	Sp	Acc	MCC
Binary	0.598 (0.18)	0.800 (0.00)	0.699 (0.02)	0.407 (0.12)
AAindex	0.571 (0.19)	0.809 (0.01)	0.690 (0.07)	0.391 (0.17)
CKSAAP	0.829 (0.09)	0.809 (0.00)	0.819 (0.09)	0.639 (0.14)
K-mer	0.811 (0.06)	0.808 (0.00)	0.810 (0.11)	0.619 (0.08)

Values in parentheses represent the standard error (SE).

**Table 3 ijms-22-02704-t003:** Prediction performance of the single encoding-employing models without any feature selection on the independent dataset.

Encoding Scheme	Sn	Sp	Acc	MCC
Binary	0.384	0.806	0.736	0.170
AAindex	0.397	0.800	0.733	0.174
CKSAAP	0.458	0.800	0.743	0.224
K-mer	0.480	0.800	0.747	0.242

**Table 4 ijms-22-02704-t004:** Prediction performance of the single encoding-employing models with feature selection on the independent dataset.

Encoding Scheme	Sn	Sp	Acc	MCC
Binary	0.445	0.801	0.742	0.214
AAindex	0.438	0.801	0.741	0.209
CKSAAP	0.504	0.805	0.755	0.268
K-mer	0.532	0.804	0.758	0.288

**Table 5 ijms-22-02704-t005:** Prediction performance of the PredNTS with feature selection.

Dataset	Predictor	AUC
Training dataset	Binary + AAindex + CKSAAP + K-mer	0.910
Independent dataset	Binary + AAindex + CKSAAP + K-mer	0.860

The weight coefficients with respect to the four types (binary, AAindex, CKSAAP, and K-mer) of the single encoding-employing models are 0.01, 0.01, 0.3, and 0.68, respectively.

**Table 6 ijms-22-02704-t006:** Performance comparison of the PredNTS with the three existing predictors on the independent dataset.

Encoding Scheme	Sn	Sp	Acc	MCC
GPS-YNO2	0.334	0.801	0.724	0.122
DeepNitro	0.339	0.803	0.726	0.128
NTyroSite	0.440	0.793	0.744	0.196
PredNTS	0.522	0.809	0.761	0.286

## Data Availability

All the data are available at http://kurata14.bio.kyutech.ac.jp/PredNTS/.

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
