# Peer review of "PredNTS: Improved and Robust Prediction of Nitrotyrosine Sites by Integrating Multiple Sequence Features"

_ijms, 2021, doi:10.3390/ijms22052704_

Round 1
Reviewer 1 Report
Overview
Nilamyani et al. present a novel machine learning (ML) method (PreNTS) for predicting tyrosine sites that are susceptible to nitration from amino acid sequences. The ML method consists of a random forest model trained on four encoding schemes for amino acid sequence; namely a binary encoding scheme, a physiochemical property encoding scheme, a scheme for encoding pairs of amino acids separated by a specified gap lengths, and an encoding of K-mers. The method is trained on a dataset of ~1000 samples, with a resulting prediction accuracy of 76.1%. The authors compare these results to three other methods for predicting nitrotyrosine sites and demonstrate state-of-the-art sensitivity, specificity, accuracy, and Matthews correlation coefficient. The authors created a web server that can be used by non-experts to predict nitrotyrosine sites from an amino acid sequence, which would be of great service to the community.
Major Comments
The reviewer has four major concerns with the work presented in the paper that need to be clarified:
-
The authors train the model on a relatively small set of training data. Based on the reviewer’s reading of the manuscript, the model is trained on ~2,400 amino acid sequences. However, the feature vectors fed into the model can have up to 8,020 rows. This presents the possibility of a model that is overfit. While the authors do indicate that features selection is performed to remove non-essential features, they do not report the final feature list or the total number of features in the final model. This makes assessing the rigor of the training impossible.
-
In the interest of reproducibility, the authors should present a full listing of features that are used in the final PreNTS model.
-
The authors indicate that 5-fold cross validation was performed. Generally, when cross validation is performed, model performance is reported as the mean +/- uncertainty (standard deviation or standard error). The results in Tables 1, 2, 3, 4, and 6, are presented without uncertainties. This makes comparison between encoding schemes and between methods impossible. Uncertainties would need to be specified to definitely demonstrate improved performance of PreNTS over other methods.
-
The manuscript needs significant grammatical editing. At many locations throughout the manuscript the meaning of sentences is unclear or ambiguous due to poor grammar. This made reading the manuscript difficult and left the reviewer uncertain about the methods, intent, and results of the paper.
Author Response
(Reviewer 1)
Comments and Suggestions for Authors
Overview
Nilamyani et al. present a novel machine learning (ML) method (PreNTS) for predicting tyrosine sites that are susceptible to nitration from amino acid sequences. The ML method consists of a random forest model trained on four encoding schemes for amino acid sequence; namely a binary encoding scheme, a physiochemical property encoding scheme, a scheme for encoding pairs of amino acids separated by a specified gap lengths, and an encoding of K-mers. The method is trained on a dataset of ~1000 samples, with a resulting prediction accuracy of 76.1%. The authors compare these results to three other methods for predicting nitrotyrosine sites and demonstrate state-of-the-art sensitivity, specificity, accuracy, and Matthews correlation coefficient. The authors created a web server that can be used by non-experts to predict nitrotyrosine sites from an amino acid sequence, which would be of great service to the community.
Major Comments
The reviewer has four major concerns with the work presented in the paper that need to be clarified:
The authors train the model on a relatively small set of training data. Based on the reviewer’s reading of the manuscript, the model is trained on ~2,400 amino acid sequences. However, the feature vectors fed into the model can have up to 8,020 rows. This presents the possibility of a model that is overfit. While the authors do indicate that features selection is performed to remove non-essential features, they do not report the final feature list or the total number of features in the final model. This makes assessing the rigor of the training impossible.
Response: Basically, we encoded a sequence window with a 41 length by the K-mer scheme, generating 8020D features, which are high-dimensional, partially redundant feature vectors. To delete the redundant features, we have used the RFE feature selection algorithm. We reported the feature selection results in Figure S1 and page 5. We have also added the final feature list in the revised version at http://kurata14.bio.kyutech.ac.jp/PredNTS/download_file/Final-Fea-PUP-Fuse.zip
In the interest of reproducibility, the authors should present a full listing of features that are used in the final PreNTS model.
Response: We have added a full list of features at http://kurata14.bio.kyutech.ac.jp/PredNTS/download_file/Final-Fea-PUP-Fuse.zip
The authors indicate that 5-fold cross validation was performed. Generally, when cross validation is performed, model performance is reported as the mean +/- uncertainty (standard deviation or standard error). The results in Tables 1, 2, 3, 4, and 6, are presented without uncertainties. This makes comparison between encoding schemes and between methods impossible. Uncertainties would need to be specified to definitely demonstrate improved performance of PreNTS over other methods.
Response: We have added the standard error to Table 1 and Table 2. Since Tables 3, 4, and 6 showed the performance on the independent datasets, we did not measure the standard errors. They are not related to uncertainties.
The manuscript needs significant grammatical editing. At many locations throughout the manuscript the meaning of sentences is unclear or ambiguous due to poor grammar. This made reading the manuscript difficult and left the reviewer uncertain about the methods, intent, and results of the paper.
Response: We have improved our article significantly.
Reviewer 2 Report
From my point a view, the basic principles needed for constructing the model and for testing it were respected, I like very much Figure 1, is very suggestive. The accuracy of predictions they specify is also good.
I have tried to use the online facility they propose and it was Ok.
It is true that identification of potential nitrotyrosine sites is not of a large interest.
There are some parts that are missing:
Author Contributions:
Funding: 275
Informed Consent Statement:
Conflicts of Interest:
Author Response
(Reviewer 2)
Comments and Suggestions for Authors
There are some parts that are missing:
Author Contributions:
Funding: 275
Informed Consent Statement:
Conflicts of Interest:
Response: We have added the above information to the revised version.
Round 2
Reviewer 1 Report
The authors have addressed this reviewer's concerns.
The reviewer identified several areas where grammar could be improved:
- Introduction, paragraph 3: "Although the experimental verified" -> "Although the number of experimentally verified"
- Introduction, paragraph 4:
- "a few of predictors have been to identify" -> "few predictors have been created to identify"
- "The iNitro-Tyr developed based on" -> "The iNitro-Tyr was developed based on"
- "A user friendly web-server was developed at" -> "A user friendly web-server was developed and is available at"
- Conclusions: "The employed RF algorithm was demonstrated be" -> The employed RF algorithm was demonstrated to be"
Author Response
The reviewer identified several areas where grammar could be improved:
Introduction, paragraph 3: "Although the experimental verified" -> "Although the number of experimentally verified"
Introduction, paragraph 4: "a few of predictors have been to identify" -> "few predictors have been created to identify"
"The iNitro-Tyr developed based on" -> "The iNitro-Tyr was developed based on"
"A user friendly web-server was developed at" -> "A user friendly web-server was developed and is available at"
Conclusions: "The employed RF algorithm was demonstrated be" -> The employed RF algorithm was demonstrated to be"
Reply: Thank you. I corrected them.